# The Role of c-MET as a Biomarker in Patients with Bladder Cancer Treated with Radical Chemo-Radiotherapy

Hélène Houssiau [1], Géraldine Pairet [2], Hélène Dano [3] and Emmanuel Seront [1,*]

1 Department of Medical Oncology, Groupe Jolimont, 7100 Haine Saint Paul, Belgium; helene.houssiau@student.uclouvain.be
2 Department of Pathology, Groupe Jolimont, 7100 Haine Saint Paul, Belgium; geraldine.pairet@helora.be
3 Department of Pathology, Cliniques Universitaires Saint Luc, 1200 Brussels, Belgium; helene.dano@saintluc.uclouvain.be
* Correspondence: emmanuel.seront@saintluc.uclouvain.be

**Abstract:** Background: Bladder cancer is a highly aggressive cancer, and muscle invasive urothelial carcinoma (MIUC) requires aggressive strategy. Concomitant chemo-radiotherapy (CRT) appears as a therapeutic option that allows bladder sparing. No biomarker is currently available to optimally select patients for CRT. Methods: We retrospectively enrolled patients with MIUC who were treated in a curative setting with CRT. Based on c-MET expression in pre-treatment tumor tissue, patients were stratified into two groups: no expression of c-MET (group A) and expression of c-MET (group B). We evaluated the outcome of these patients based on c-MET expression. Results: After a median follow-up of 40 months, 13 patients were enrolled in this analysis, 8 in group A and 5 in group B. The disease recurrence was 25% in group A and 100% in group B. Compared to group A, patients from group B experienced more frequent and more rapid recurrence in terms of metastases; the 3-year metastatic recurrence rate was 13% and 100%, respectively. The c-MET expression was also associated with a higher rate of cancer-related deaths. Conclusions: In this retrospective analysis, c-MET expression was associated with worse disease-free survival and survival in patients treated radically with CRT.

**Keywords:** c-MET; bladder cancer; radiotherapy; chemotherapy; metastasis-free survival; biomarker

## 1. Introduction

With more than 350,000 newly diagnosed cases and approximately 150,000 deaths each year worldwide, bladder cancer is the seventh most common cancer in the world; urothelial carcinoma (UC) accounts for 90% of bladder cancer [1,2]. The majority of patients (75%) present with localized and non-muscle invasive UC (NMIUC) and are treated with surgical resection, intravesical chemotherapy and/or intravesical injection of bacillus Calmette–Guerin (BCG) [3]. In cases of muscle involvement (MIUC), standard treatment includes neoadjuvant chemotherapy followed by surgery or concomitant chemotherapy plus radiotherapy (CRT). Platinum-based chemotherapy remains the cornerstone cytotoxic agent in the neoadjuvant setting. In the VESPER phase III trial, cisplatin plus gemcitabine (CG) was compared to dose dense methotrexate/vinblastine/adriamycin/cisplatin (dd-M-VAC); the pathological complete response rate (pCR) reached 42% vs. 36% in the dd-MVAC compared to CG, respectively [4]. Whether reaching the pCR with neoadjuvant chemotherapy requires further cystectomy remains unknown. Another curative option is CRT; even if no trial has prospectively compared these two strategies, CRT appears as a promising treatment, allowing organ preservation and preventing aggressive surgery in frail and/or potential micro-metastatic patients. The choice of chemotherapy in this setting includes 5-fluorouracyl (5FU) + mitomycin. James et al. reported in a large phase III trial comparing CRT vs. RT alone that locoregional disease-free survival was significantly better in the CRT group than in the RT group, with 2-year recurrence-free rates of 67% versus 54%,

respectively. There was also a trend for better overall survival (OS) with CRT compared to RT alone [5]. Other regimens include weekly cisplatin or weekly gemcitabine [6]. The success of this strategy mainly depends on the correct selection of patients: ideally T2, no hydronephrosis, no carcinoma in situ (Cis), maximal tumor resection on TURBT, unifocal tumor and good bladder function/capacity [7]. Currently, no biomarker exists concerning the prediction of efficacy when considering neoadjuvant chemotherapy followed by surgery or CRT strategy. c-MET is a tyrosine kinase receptor that, after binding to hepatocyte growth factor, activates different signaling pathways involved in cell survival, proliferation and angiogenesis. Excessive and uncontrolled activation of c-MET has been shown to trigger development and progression of multiple cancers. c-MET has also been shown to be involved in resistance mechanisms to treatments such as chemotherapy and radiotherapy [8]. In a retrospective study, we evaluated the outcome of patients treated with CRT for a bladder cancer based on c-MET expression.

## 2. Materials and Methods

We retrospectively enrolled in this study patients with confirmed MIUC of the bladder who were treated with CRT in a curative intent from January 2016 to January 2021 in Jolimont Hospital. This retrospective analysis was performed in accordance with the Declaration of Helsinki and approved by the Ethics Committee of Groupe Jolimont.

To be enrolled in this analysis, patients had to have urothelial carcinoma < cT3 based on TNM staging (mixed histology with predominant UC were allowed), with no Cis, no hydronephrosis and no renal failure. No distant metastasis or lymph node had to be seen on thoraco-abdominal CT. Maximal resection on TURBT was mandatory. Patients were allowed to receive weekly cisplatin, weekly gemcitabine or 5FU + mitomycin. They could not have received neoadjuvant chemotherapy or radiotherapy alone.

Tissue expression of c-MET was analyzed with immunohistochemistry (IHC) using a c-MET antibody (rabbit monoclonal antibody (SP44), Ventana Medical Systems). Both cytoplasmic and membranous staining were considered positive. Scoring of the c-MET staining was as follows: 0 if no staining; 1+ if weak staining in any amount of tumor cells and moderate staining in <50% of tumor cells; 2+ if ≥50% of tumor cells showed incomplete membranous and/or cytoplasmic staining with at least moderate intensity but <50% cells with strong intensity; and 3+ if ≥50% of tumor cells with membranous and/or cytoplasmic staining with strong intensity. Patients were divided in two groups: group A = patient with no IHC staining (0), and group B = patients with IHC staining (1+, 2+ or 3+) (Figure 1).

Disease-free survival (DFS) was calculated from the start of CRT to the date of local NMIUC (Ta, T1, Cis), local MIUC (≥T2) or distant metastasis (including regional lymph nodes) recurrence. This disease recurrence rate at 2 and 3 years was evaluated. OS time was calculated from the time of diagnosis to the date of death or last follow-up. Kaplan–Meier analysis was used to estimate the DFS, metastases-free survival (MFS) and OS. In case of recurrence or progression, analysis of c-MET expression was assessed on available tumor tissue samples.

**c-MET tissue expression**

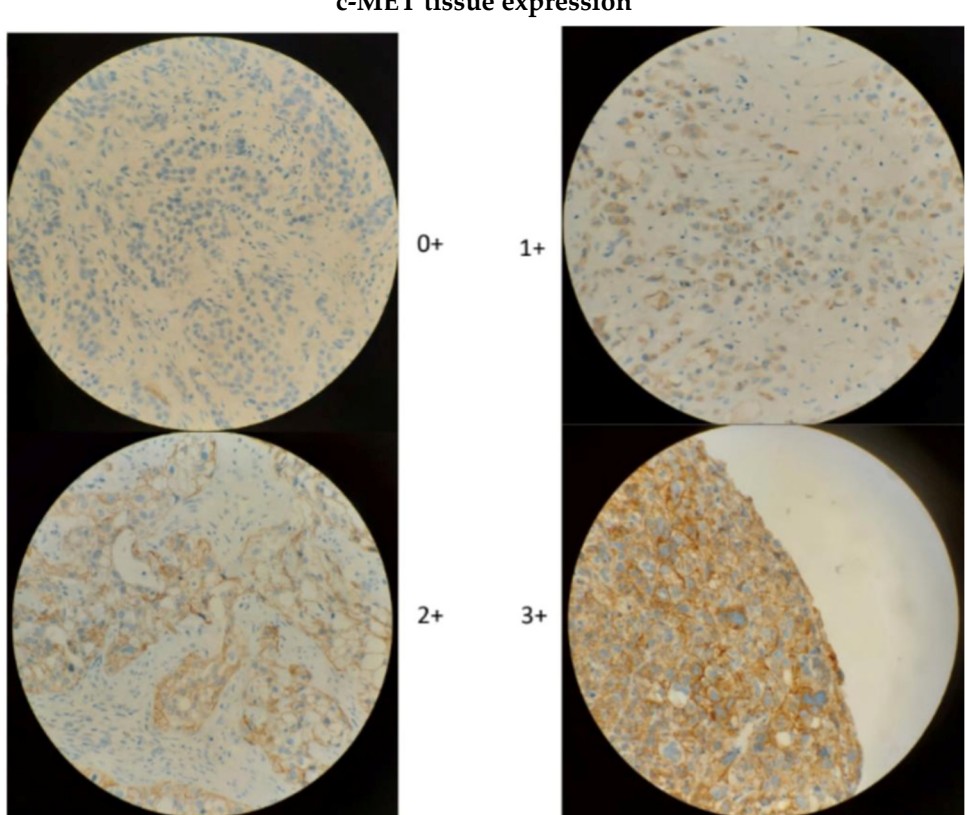

**Figure 1.** c-MET expression in immunohistochemistry staining (0, 1+, 2+, 3+).

### 3. Results

Thirteen patients corresponding to the inclusion criteria were treated with CRT for MIUC in the defined period of time. The median follow-up was 40 months, ranging from 18 to 60 months. The baseline characteristics are described in Table 1. Eight patients presented with no c-MET expression (group A), and five patients presented with c-MET expression (group B).

**Table 1.** Baseline characteristics.

|  | c-MET 0/1+ N (%) | c-MET 2/3+ N (%) |
|---|---|---|
| N | 8 | 5 |
| Female/Male | 1 (13)/7 (87) | 0 (0)/5 (100) |
| Age, Median (Range) | 70 (60–78) | 67 (64–78) |
| Previous history of UC (pTis, pT1) | 4 (50) | 1 (20) |
| Previous BCG therapy | 4 (50) | 0 (0) |
| Grade Low Intermediate High | 0 (0) 5 (62) 3 (38) | 0 (0) 1 (20) 4 (80) |
| Chemotherapy regimen 5FU + mitomycin Cisplatin weekly Gemcitabine | 3 (38) 3 (38) 2 (24) | 2 (40) 3 (60) 0 (0) |

Two patients in group A and five patients in group B experienced, after CRT, disease recurrence (NMIUC, MIUC and/or metastatic events) during the follow-up, representing a disease recurrence rate of 25% and 100%, respectively. The 2- and 3-year recurrence rate in group A was 13% and 13%, respectively, and in group B, it was 60% and 100%, respectively. The median DFS was 56 months in group A and 23 months in group B ($p$ = 0.001; HR 0.1, 95% CI 0.01–0.6).

NMIUC recurrence occurred in two patients from group A (25%), both occurring after 2 years. One patient from group B (20%) developed NMIUC within the 2 years.

No MIUC recurrence was observed in group A (0%), and one patient from group B (20%) experienced an MIUC recurrence, occurring within the 3 years.

In terms of metastatic resurgence, one patient (13%) from group A and five patients (100%) from group B experienced disease recurrence outside the bladder (lymph nodes or distant lesions). The patient in the group A presented with metastatic recurrence within 2 years, while two patients from group B were diagnosed with metastatic lesions within 2 years and five within 3 years; the 2- and 3-year metastatic recurrence rate was 13% and 13%, respectively, in group A and 40% and 100% in group B, respectively. The median MFS was better in group A compared to group B (not reached vs. 25 months, respectively; $p$ = 0.009, HR 0.08, 95% CI 0.01–0.4) (Figure 2). In the five patients with c-MET expression, we showed that patients with only a score of 1+ experienced a better outcome compared to patients with expression scores of 2/3+ (Figure 2).

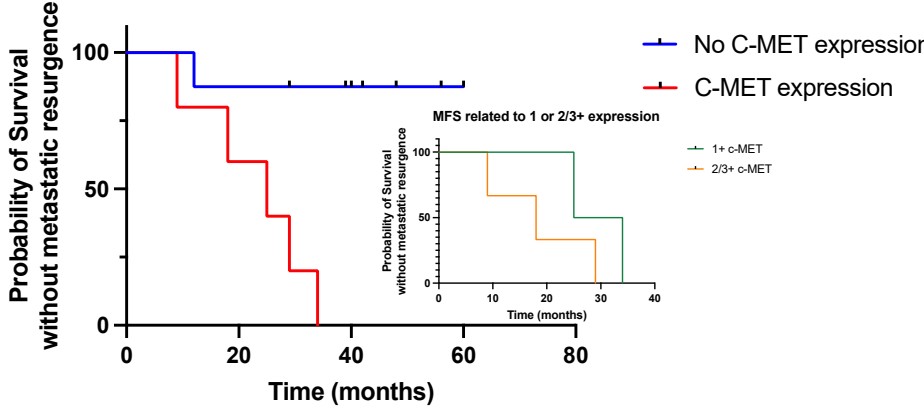

**Figure 2.** Metastasis-free survival based on c-MET expression.

Metastatic recurrence manifested in group A as liver metastases (n = 1) and as lymph nodes (n = 2), liver metastases (n = 2), bone metastases (n = 1) in group B.

The median survival was 56 months in group A and 48 months in group B ($p$ = 0.08; HR 0.3, 95% CI 0.04–2). There was only one disease-related death in group A (13%) compared to three in group B (60%).

Cystectomy was not performed on any patients from group A, while in group B, it was performed on one patient due to resurgence of MIUC.

At recurrence, c-MET expression was assessed in available tumor samples. In group A, one NMIUC sample (among the two NMIUC recurrences) had a 2+ c-MET expression score, and one metastatic lesion was moderately/strongly positive for c-MET. In group B, all available lesions were c-MET-positive (one NMIUC, one MIUC sample and two visceral metastatic lesions were available for tissue analysis).

## 4. Discussion

c-MET is an important tyrosine kinase receptor that plays a key role in cell growth, proliferation and survival through the activation of various survival pathways, including the PI3K-AKT-mTOR cascade, the Ras-Raf-MEK cascade and the STAT3 pathway [9]. Aberrant c-MET expression has been reported in 25–30% of invasive UC and is associated

with a poor prognosis [10]. In bladder UC, overexpression of c-MET has been shown to be associated with higher histological grade and stage, as well as with disease progression, and shortened metastasis-free and overall survival [11–14]. Preclinical studies have shown that downregulation of c-MET inhibits proliferation, induces cell apoptosis and suppresses cell motility in UC, confirming the role of this tyrosine kinase receptor in UC carcinogenesis and progression. Furthermore, c-MET has also been shown to be involved in resistance to cisplatin chemotherapy [14].

To our knowledge, this is the first time that c-MET expression has been shown to be associated with the aggressiveness and poorer outcomes of bladder cancer patients treated with CRT.

First, despite an aggressive radical treatment, metastatic progression was shown to occur at a higher frequency in patients with c-MET expression (group B) compared to patient with no c-MET expression (group A). Furthermore, these recurrences occurred more rapidly in group B compared to group A, with a higher percentage of metastatic recurrences at 3 years (100% vs. 13%, respectively).

In terms of survival, there was only one disease-related death in group A, while there was three in group B (13% vs. 60%, respectively). Two main hypotheses emerge, (1) consistent with previous reports, c-MET expression may be associated with the aggressiveness of the disease and early dissemination, explaining why a local CRT strategy is not appropriate in these patients with micro-metastases, as chemotherapy doses are only radiosensitive with no intense systemic action; (2) c-MET may be associated with resistance to CRT, and these clones may develop despite an aggressive local therapy, leading to local recurrence and systemic dissemination. Despite a very low number of patients, we evaluated the impact of c-MET expression on MFS, and we showed that scores of 2/3+ were associated with poorer outcomes compared to 1+ expression scores.

It is difficult to draw any conclusions from this study and assess the role of c-MET as a potential biomarker when considering CRT for bladder cancer. These results should not be compared to other CRT trials, as there is a very low number of patients, the endpoints are not similar, and a longer follow-up is clearly required in our study. Our results may just suggest that expression of c-MET is associated with poorer outcomes (early metastatic recurrence and decreased survival) when considering CRT. A sooner follow-up should be proposed in these patients, with regular cystoscopy and systemic imaging. Whether a positron emission tomography could help in excluding infra-centimetric metabolic lesions to improve patient selection remains unknown. Whether systemic chemotherapy before local treatment is an appropriate strategy also remains unclear. Further clinical studies with prospective evaluation of the role of c-MET are needed to clarify its role as biomarker.

Although recommended, CRT is not often proposed by physicians, which explains the low number of patients in this trial and a potential issue when conducting further trials without an appropriate biomarker. It is important of course to correctly select patients who will not experience rapid systemic progression based on histopathological features of the cancer.

There are many limitations in this study, including the retrospective analysis, the very limited number of patients, the variability in pre-treatment imaging, the treatment heterogeneity, the imbalance between the two groups' characteristics, as well as the non-standardized way to assess c-MET. Another important limitation is the fact that we did not assess the role of c-MET expression in a cohort of patients treated with neoadjuvant chemotherapy followed by surgery. However, this opens the way for the identification of a biomarker, including c-MET, in the choice of treatment for patients with MIUC.

## 5. Conclusions

Despite multiple limitations, including the low number of patients, this study highlights the importance of identifying potential biomarkers to predict the efficacy or inefficacy of treatments. In this retrospective study, c-MET was associated with poorer outcomes and shorter metastasis-free survival. This should be confirmed in further prospective studies.

**Author Contributions:** Conceptualization, H.H. and E.S.; methodology, H.D., G.P., H.H. and E.S.; formal analysis, H.H. and E.S.; investigation, H.D., G.P., H.H. and E.S.; data curation, H.H. and E.S.; writing—original draft preparation, H.H.; writing—review and editing, H.H. and E.S.; visualization, H.H. and E.S.; supervision, E.S. All authors have read and agreed to the published version of the manuscript.

**Funding:** This research received no external funding.

**Institutional Review Board Statement:** The study was conducted in accordance with the Declaration of Helsinki and approved by the Ethics Committee of Groupe Jolimont, approval number: 150923 (15 September 2023).

**Informed Consent Statement:** Not applicable.

**Data Availability Statement:** The data presented in this study are available n request from the corresponding author.

**Conflicts of Interest:** The authors declare no conflict of interest.

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
