# Peer review of "The Role of c-MET as a Biomarker in Patients with Bladder Cancer Treated with Radical Chemo-Radiotherapy"

_curroncol, doi:10.3390/curroncol30120770_

Round 1
Reviewer 1 Report
Comments and Suggestions for Authors
In the article "The role of c-MET as biomarker in patients with bladder cancer treated with radical chemo-radiotherapy" Hélène Houssiau addresses a very interesting topic that can certainly be translated beyond bladder cancer treated with chemoradiotherapy. Pretreatment expression of c -MET demonstrates a significant prognostic value even if the data must be interpreted in the context of a small study group. The interesting thing is the discussion between the 1+ and 2/3+ scores for MFS. (I mention that all abbreviations must be explained). The idea ": consistently with previous reports, c-MET expression may be associated with aggressiveness of the disease and early dissemination, explaining why a local CRT strategy is not appropriate in these patients with micro-metastases" should be detailed... however, we are discussing chemo-radiotherapy, so we address and micrometastases inevitably, even if the main role of chemotherapy is radiosensitization. I would prefer to express the idea that I would consider a "more intense systemic therapy regimen"
Author Response
Thank you for your pertinent observation. We have adapted the text according to your recommendations. “1) consistently with previous reports, c-MET expression may be associated with aggressiveness of the disease and early dissemination, explaining why a local CRT strategy is not appropriate in these patients with micro-metastases as chemotherapy doses are only radiosensitizer with no intense systemic action; »
Reviewer 2 Report
Comments and Suggestions for Authors
In the paper named “The role of c-MET as biomarker in patients with bladder cancer treated with radical chemo-radiotherapy” author make a retrospective study to evaluate the outcome of patients based on c-MET expression. This is an interesting work take into account that until know no biomarkers are available to select optimally patients to received concomitant Chemo-radiotherapy.
The main citizen to this word is the low number of patients enrolled in the study, only 13 patients.
Other minor questions are:
1) The patient inclusion criteria are well defined, only one question refered to the IHC. I think that some image showing the positive and the negative staining will be recommended
2) In table 1 the data of the grade give number of patients that do not mach with the patients per group. For example c-MET 0/1+ in this group there are N=8 however in the grade there are 8 intermediate and 5 with high grade.
Author Response
The paper named “The role of c-MET as biomarker in patients with bladder cancer treated with radical chemo-radiotherapy” author make a retrospective study to evaluate the outcome of patients based on c-MET expression. This is an interesting work take into account that until know no biomarkers are available to select optimally patients to received concomitant Chemo-radiotherapy.
The main citizen to this word is the low number of patients enrolled in the study, only 13 patients.
Thank you for your feedback,
We also regret the small number of patients in this study, but given that the indications for treatment with radio-chemotherapy remain limited at this time, it is still acceptable. As you point out, we hope one day to see the emergence of biomarkers that can help us in our therapeutic choices and extend this treatment to a larger number of patients. We, of course, highlighted this limitation in the discussion and conclusion.
Other minor questions are:
1) The patient inclusion criteria are well defined, only one question refered to the IHC. I think that some image showing the positive and the negative staining will be recommended
Thank you for this very pertinent comment, we have added a photo with the IHC results. (Figure 1)
2) In table 1 the data of the grade give number of patients that do not mach with the patients per group. For example c-MET 0/1+ in this group there are N=8 however in the grade there are 8 intermediate and 5 with high grade.
Thank you for pointing out this mistake, the table has been modified.
Reviewer 3 Report
Comments and Suggestions for Authors
Please provide more detail regarding scoring of IHC. Was it done blindly from clinical outcomes? Only one pathologist?
Author Response
Please provide more detail regarding scoring of IHC. Was it done blindly from clinical outcomes? Only one pathologist?
Thank you for your pertinent question.
The two pathologists who read the IHC results did not have access to the from clinical outcomes and had an independent view on the IHC.